# BACE-1 and γ-Secretase as Therapeutic Targets for Alzheimer’s Disease

**DOI:** 10.3390/ph12010041

**Published:** 2019-03-19

**Authors:** Miguel A. Maia, Emília Sousa

**Affiliations:** 1Laboratory of Organic and Pharmaceutical Chemistry, Department of Chemical Sciences, Faculty of Pharmacy, University of Porto, Rua Jorge Viterbo Ferreira, 228, 4050-313 Porto, Portugal; miguelmaia2@gmail.com; 2Interdisciplinary Centre of Marine and Environmental Research (CIIMAR), Terminal de Cruzeiros do Porto de Leixões, Av. General Norton de Matos s/n, 4450-208 Matosinhos, Portugal

**Keywords:** Alzheimer’s disease, amyloid hypothesis, γ-secretase, BACE-1

## Abstract

Alzheimer’s disease (AD) is a growing global health concern with a massive impact on affected individuals and society. Despite the considerable advances achieved in the understanding of AD pathogenesis, researchers have not been successful in fully identifying the mechanisms involved in disease progression. The amyloid hypothesis, currently the prevalent theory for AD, defends the deposition of β-amyloid protein (Aβ) aggregates as the trigger of a series of events leading to neuronal dysfunction and dementia. Hence, several research and development (R&D) programs have been led by the pharmaceutical industry in an effort to discover effective and safety anti-amyloid agents as disease modifying agents for AD. Among 19 drug candidates identified in the AD pipeline, nine have their mechanism of action centered in the activity of β or γ-secretase proteases, covering almost 50% of the identified agents. These drug candidates must fulfill the general rigid prerequisites for a drug aimed for central nervous system (CNS) penetration and selectivity toward different aspartyl proteases. This review presents the classes of γ-secretase and beta-site APP cleaving enzyme 1 (BACE-1) inhibitors under development, highlighting their structure-activity relationship, among other physical-chemistry aspects important for the successful development of new anti-AD pharmacological agents.

## 1. Introduction

### 1.1. Alzheimer’s Disease–Epidemiology

AD is recognized by the World Health Organization (WHO) as a global public health priority [1]. Notwithstanding the advances in the understanding of AD pathogenesis since Alois Alzheimer reported the first case in 1907 [2], there are still a lot of uncertainties regarding the mechanisms involved in disease progression. AD is the most prevalent cause of dementia-acquired progressive cognitive impairment sufficient to impact on activities of daily living, being a major cause of dependence, disability and mortality. The WHO estimated that in 2010, 35.6 million worldwide were living with dementia [3]. This figure is projected to almost double every 20 years, reaching 65.7 million by 2030 and 115.4 million by 2050. Europe, with an estimated 10 million cases of dementia in 2010 and acquiring progressively an old population structure [4], has a projected increase to 14 million cases in 2030 [1].

In the United States of America (USA), the total annual payments for health care, long-term care and hospice care for people with AD or other dementias are projected to increase from $259 billion in 2017 to more than $1.1 trillion in 2050 [5]. These numbers show the dramatic impact that AD and the other dementias will have in the health care systems in the future.

### 1.2. Pathology

The main characteristics of Alzheimer’s pathology are the presence of amyloid plaques and neurofibrillary tangles (aggregates of hyperphosphorylated tau protein) [5]. In addition, neuropil treads, dystrophic neurites, associated astrogliosis, and microglial activation frequently coexist. The downstream consequences of these pathological processes include neurodegeneration with synaptic and neuronal loss leading to macroscopic atrophy [6]. Research suggests that the brain changes associated with AD may begin 20 or more years before symptoms appear [7,8]. When the initial changes occur, the brain compensates for them, enabling individuals to continue to function normally. As neuronal damage increases, the brain can no longer compensate for the changes and individuals show subtle cognitive decline. Later, neuronal damage is so significant that individuals show obvious cognitive decline, including symptoms such as memory loss or confusion as to time or place [5].

### 1.3. Current Pharmacology and Drug Development

Currently, five drugs are approved for the treatment of AD (Table 1). These include four acetylcholinesterase inhibitors (AChEi) - tacrine (approved in 1993), donepezil (approved in 1996), rivastigmine (1998), and galantamine (approved in 2001) [9]. This class of drugs act by blocking the process that downregulate the neurotransmitter acetylcholine (ACh), a key signaling agent for nerve cells communication. Although tacrine has been the first AChEi approved by Food and Drug Administration (FDA), it is currently not used due to its hepatotoxicity [10]. Donepezil, rivastigmine, and galantamine belong to the same therapeutic class, but present different pharmacodynamic (PD) and pharmacokinetic (PK) profiles: donepezil is a noncompetitive reversible AChEi; galantamine is a selective reversible AChEi and a positive allosteric modulator of nicotinic ACh receptors; and rivastigmine acts as an inhibitor of both acetylcholinesterase (AChE) and butyrylcholinesterase (BuChE) [11].

An alternative therapy to AChEi is the administration of memantine, a non-competitive antagonist of *N*-methyl-D-aspartate (NMDA) receptor, approved in 2004. This drug regulates the activity of glutamate in the brain. Attachment of the neurotransmitter glutamate to the cell surface of NMDA receptors allows calcium to enter the cell. This is an important event in cell signaling, as well as in learning and memory. However, in AD patients, excess glutamate released from damaged cells leads to overexposure to calcium and accelerates cell damage. Memantine disrupts this chain of events by blocking the NMDA receptors [11,12]. In Table 1 are comprised the current drugs used in the therapy of AD.

### 1.4. Disease-Modifying Therapies and Drug Development

The drugs currently available for the treatment of AD only relieve symptoms, not being able to impact in the progression of the disease. There is an urgent need for the development of disease modifying therapies or treatments (DMT’s) able to prevent, delay, or slow the progression and target the primary pathophysiology mechanisms of AD. It has been estimated that the overall frequency of the disease would be decreased by 40% if the onset of the disease could be delayed by 5 years [13].

Unfortunately, drug development for AD has proven to be very difficult. The high cost of drug development, the relatively long time needed to observe whether an investigational treatment affects disease progression and the needed capacity of the drug to cross the blood-brain barrier (BBB) are some factors that contribute to the absence of new approved drugs for AD. Even considering the compounds that reach clinical trials, many of them fail to prove efficacy or present unacceptable toxicity, both small molecules and immunotherapies [9]. Considering the decade between 2002 and 2012, 244 compounds were assessed in 413 trials for AD. Of the agents which advanced to phase III, only one was advanced to FDA and approved for marketing (memantine). This represents an overall success rate for approval of 0.4%, which is among the lowest for any therapeutic area [9].

However, besides the mentioned difficulties in the development of new therapies for AD, there are currently large research programs for the drug development of new anti-AD agents. In 2018, there were 112 agents in the AD pipeline, of which 23 in phase I, 63 in phase II and 26 in phase III. Across all stages, 63% are DMTs, 34% symptomatic agents for neuropsychiatric and behavioral changes, and 3% have undisclosed mechanism of action (MoA) [14]

Figure 1 summarizes the MoA of the agents in clinical trials in 2018. Twenty five percent of the agents have an amyloid-related MoA (12% immunotherapy and 14% small molecules), 14% anti-tau, 22% neurotransmitter-based (symptomatic treatment agents), 18% neuroprotection/ anti-inflammatory, 18% metabolic and 3% with other mechanisms [14].

As depicted in Figure 1, Aβ appears as the main therapeutic target for AD, with several pharma/biopharma industries trying to develop agents (as immunotherapies or small molecules) able to decrease Aβ accumulation/aggregation and to show a positive effect on AD pathology.

## 2. The Amyloid Hypothesis of AD

The amyloid hypothesis, the current prevalent theory of AD pathogenesis, suggests that the accumulation of pathological forms of Aβ, due to an increase of production and/or decreased clearance, is the primary pathological process in AD [3,15]. The accumulation of Aβ peptides leads to their oligomerization and formation of Aβ plaques. These plaques generate an anti-inflammatory response causing oxidative stress in neurons and disrupting normal kinase and phosphatase activity, resulting in hyperphosphorylation of tau protein and subsequent neurofibrillary tangle formation. This cascade of events leads to abnormal signaling and synaptic impairment, resulting ultimately in neuronal dead and dementia in the AD patient [16]. This is schematized in Figure 2.

Aβ is produced by a two-step sequential cleavage of amyloid precursor protein (APP): first beta-site APP-cleaving enzyme (BACE-1) cleaves APP to generate soluble APPβ (sAPPβ) and a 99 amino acid fragment (C99), which then suffers several cleavage events by γ-secretase to produce peptides of different lengths from 38 to 43 amino acids, being Aβ40 and Aβ42 the main products and both playing a key role in the aggregation of neuritic plaques [15,17] (Figure 3). The longer peptide Aβ42 has been shown to be the most prone to aggregation, with an increased Aβ42:Aβ40 ratio observed in the familial form of the disease [17,18].

The activity of β-cleavage is considered to determine the total amount of Aβ production, and the efficiency of the successive γ-cleavage impacts the production ratio of toxic Aβ42 to total Aβ [15].

The amyloid hypothesis is strongly supported by genetic findings. All the known familial Alzheimer’s disease (fAD) mutations are involved in either Aβ increase generation or processing and result in relative overproduction of toxic forms of Aβ, namely Aβ42 [3]. On the other hand, it was reported that a mutation in the APP gene (A673T), identified from a set of whole-genome sequence data of 1795 Iceland people, results in a lifelong decrease in APP cleavage by β-secretase conferring a reduced clinical risk of AD and age-related decline [19,20].

Besides the fact that occurrence of AD and genetic mutations associated with Aβ formation supports the amyloid hypothesis, some observations remains without explanation, with histological studies being controversial in the correlation between the formation of Aβ plaques and cognitive decline [21].

Karran et al. [22] defend the prevention of Aβ deposition in patients that are in risk of developing AD as a robust test of the amyloid hypothesis. If individuals are at risk of developing AD but Aβ deposition has not yet occurred, then a significant reduction of Aβ42 production or a decrease in the longer amyloid-β/shorter amyloid-β ratio would be expected to prevent the onset of disease within a normal lifetime [22]. Although there are further questions concerning the exact neuronal toxicity of Aβ, the amyloid hypothesis is still broadly accepted as the general pathological cascade of events in AD [22,23].

## 3. Amyloid Targeting Strategies

Following the amyloid hypothesis, where a high brain Aβ level is observed as an important factor in AD pathogenesis, pharmacological intervention to reduce its production or improve the clearance has become a logical approach for AD therapy development.

Considering the 112 agents in the AD pipeline in 2018 [14], there are 29 with a mechanism of action amyloid related, 13 corresponding to immunotherapeutic agents (Table 2) and 16 to small molecules (Table 3) involved in phase I, II and III clinical trials.

With regard to small molecules, different targets have been identified for activity modulation hoping to get a marked impact in the amyloid cascade and in disease progression. Among 16 anti-amyloid agents identified in the AD pipeline, six have their mechanism of action centered in the activity of β or γ-secretase proteases, covering almost 40% of the identified agents.

In fact, in the last years several pharmaceutical industries and other research centers have led research programs to discovered potent compounds for the modulation or inhibition of these two targets [16,17,24,25,26]. The discovery and optimization activities which lead to the development of these compounds will be herein detailed.

### 3.1. Gamma Secretase Inhibitors and Modulators

Gamma secretase (GS) (Figure 4) is an aspartyl protease composed by a complex of four different membrane proteins: presenilin (PS), presenilin enhancer 2 (Pen- 2), nicastrin (Nct), and anterior pharynx-defective 1 (Aph-1) [27]. PS is the catalytic component of γ-secretase. In humans, PS is encoded by the PSEN1 (PS-1) gene on chromosome 14 or the PSEN2 (PS-2) gene on chromosome 1, and mutations in both genes have been found to cause fAD [28]. The products of these genes (PS-1 and PS-2) are nine transmembrane domain (TMD) proteins that form the catalytic subunit of GS [17]. GS cleaves several type-I transmembrane proteins (over 90 reported substrates), being APP and Notch the best characterized substrates [29].

The activity of GS on the substrate APP occurs after the cleavage performed by β-secretase (BACE-1). Then, GS perform a series of cleavages within the transmembranar domain of the remaining fragment (C99), termed epsilon (ε), zeta (ζ), and gamma (γ) cleavages, allowing the generation of Aβ peptides of different lengths (Figure 5).

The ε cleavage (1) releases the APP intracellular domain (AICD) and produces Aβ49 or Aβ48 [28,31]. Then, the carboxypeptidase cleavages ζ (2) and γ (3 and 4) progressively trims these longer Aβ forms in both Aβ40 and Aβ42 [31,32]. The successive cleavage events performed by GS consists in four cycles to generate Aβ40 (49-46-43-40) and Aβ38 (48-45-42-38). Further cleavage will subsequently generate the shorter isoforms Aβ39and Aβ37 [33].

Many fAD-causing mutations in PS have been found to decrease the catalytic activity of GS [34,35] with the most pronounced effect on the fourth cleavage cycle [36,37]. This loss of function contributes to the increased Aβ42:Aβ40 ratio observed in the familial form of the disease [38]. Aβ42 is consider the most toxic Aβ isoform due to its high propensity to form fibrillary and non-fibrillary aggregates. On the other hand, shorter Aβ peptides are speculated to be less toxic or even neuroprotective [33].

#### 3.1.1. γ-Secretase Inhibitors

Based on the genetic evidence of the role of GS and Aβ formation on AD, the pharmaceutical industry made efforts in developing compounds able to inhibit this protease and, consequently, reduce the amount of Aβ peptide formation. Two key γ-secretase inhibitors (GSIs), semagacestat (**1**) from Eli Lilly & Co and later avagacestat (**2**) from Bristol-Myers Squibb (Figure 6) advanced in late-stage clinical trials for AD (phase III and phase II, respectively) [39,40].

Unfortunately, due to GS high promiscuity in terms of substrates, several side effects were observed throughout the clinical trials with these compounds [25]. The main responsible pointed for these adverse effects was Notch, a single-pass type I transmembrane receptor [41]. Notch is processed in a similar way as C99, where the Notch intracellular domain (NICD) is released after the ε-cleavage and translocate to the nucleus to regulate gene expression and to mediate important intercellular communication functions as cellular differentiation. Additionally, Notch pathway is involved in neurogenesis, neuritic growth and long-term memory [41]. The inhibitory effect of GSIs blocks the ε-cleavage of γ-secretase affecting the Notch signaling, leading to several adverse events as skin cancer, decreased lymphocyte count and/or memory loss. These adverse effects were observed in phase III clinical trial of semagacestat (**1**) [40].

Consequently, subsequent drug-development programs have aimed to achieve a greater separation between APP and Notch inhibition. Bristol-Myers Squibb developed avagacestat (**2**) and reported it as having a 137-fold selectivity for APP over Notch in cell culture, and to robustly reduce cerebrospinal fluid (CSF) Aβ levels without causing Notch-related toxicity in rats and dogs [42]. However, a phase II clinical trial performed with avagacestat demonstrated a side-effect profile similar to the one found with semagacestat (**1**) [39]. These results were attributed to an overestimation of the selectivity of avagacestat between Notch and APP, with other researchers indicating an only 3-fold selectivity for cleavage of an APP substrate compared with a Notch substrate [43].

The disappointing results obtained with these two GSI led to discontinuation of the development of GSIs as therapeutic strategy against AD. Additionally, other options as GS modulators arose as safer disease modifying therapies for AD.

#### 3.1.2. Gamma secretase Modulators

A γ-secretase modulator (GSM) is defined as a compound that changes the relative proportion of the Aβ isoforms generated while maintaining the rate at which APP is processed [33].

##### Nonsteroidal Anti-inflammatory Drug Derived GSM’s

The first generation of GSMs was discovered from an epidemiological study documenting a reduced prevalence of AD among users of nonsteroidal anti-inflammatory drugs (NSAIDs) [44]. Then, in 2001 Weggen et al. [44] reported that NSAIDs (Figure 7) were able to decrease the Aβ42 peptide accompanied by an increase in the Aβ38 isoform, indicating that NSAIDs modulate γ-secretase activity without significantly perturbing other APP processing pathways or Notch cleavage [44].

This change in the cleavage pattern could be explained by (1) a decrease in the probability of releasing longer Aβ from the enzyme-substrate complex (defined as dissociation constant, κ_d_) of GS, or (2) an increase in the cleavage activity (defined by the catalytic constant, κ_cat_) of GS. Using C99 as a substrate, Chavez-Gutierrez et al. [36] showed that the Aβ40/Aβ43 and Aβ38/Aβ42 ratios were decreased by all the PS mutations studied, suggesting an deterioration of the four cleavage cycle of GS (γ’ cleavage). Additionally, using Aβ42 as a substrate, Okochi et al. [35] measured the kinetic constants for the γ cleavage and reported that GSMs decreased κ_d_ and increased κ_cat_, while PS mutations caused the opposite effect. In sum, these results suggest that the modulation effect of GSMs could be an effective approach to reverse the effect of PS mutations by restoring the normal ratios of Aβ peptides formed, and in this way modifying the disease pathology and progression [17].

Though, and as expected, the use of NSAID as GSMs present some issues in terms of gastrointestinal and renal toxicity due to its activity against cyclooxygenase 1 (COX-1), compromising its use as a long-term therapeutic solution [45]. Fortunately, COX-1 inhibition activity was shown to be independent of γ-secretase modulation activity. For instance, flurbiprofen is a COX-1 inhibitor administered in the clinic as a racemate. However, the *R*-enantiomer, tarenflurbil (**6**) (Figure 8), showed reduced activity in terms of COX-1 inhibition while maintaining its action as a GSM [45].

Tarenflurbil (**6**, Flurizan™, Myriad Genetics & Laboratories) reached phase III clinical trials. Unfortunately, results showed no difference between **6** and the placebo, the failure being attributed to the insufficient PD properties of Flurizan™, namely its inadequate capacity to penetrate the brain and engage its target at doses sufficient to yield an effect. Actually, poor CNS penetration of tarenflurbil (**6**) was previously reported in preclinical studies in rodents, with a CSF/plasma ratio of 1.3%. The unsatisfactory results obtained led to the discontinuation of tarenflurbil (**6**) clinical development [46].

Later, two NSAID carboxylic acid derivatives developed by Chiesi (CHF5074, **7**) and FORUM Pharmaceuticals (EVP-0015962, **8**) (Figure 9) were also tested in clinical trials.

Chiesi Farmaceutici developed CHF5074 (**7**) based on the scaffold of tarenflurbil (**6**). The replacement of the *R*-methyl substituent by a cyclopropyl group led to a complete removal of COX inhibition (at 100 µM for COX-1 and 300 µM for COX-2). Aβ42 production inhibition were improved with the addition of chlorine substituents on the terminal phenyl ring, allowing a 7-fold comparing with tarenflurbil (**6**) [47] (Figure 10). The carboxylic acid function was proposed to interact with a lysine residue of APP located close to the membrane interface, with the lipophilic substituents serving as membrane anchors [48].

Despite the increase in potency and selectivity accomplished with CHF5074 (**7**), no sufficient improvements were achieved in terms of PK parameters. CHF5074 (**7**) failed to demonstrate efficacy due to its poor druglike properties and extremely poor CNS penetration (brain to plasma ratio = 0.03–0.05) [49].

EVP-0015962 (**8**) was developed by FORUM Pharmaceutical (ex Envivo) through the introduction of additional substituents on the biphenylacetic acid core of *R*-flurbiprofen. This compound showed improved in vitro potency (Aβ42 IC_50_ = 67 µM) with reproducible animal efficacy, with a reduction of 22% and 38% with oral doses of 10 and 30 mg/kg, respectively [33]. However, as general characteristic of NSAID derived GSMs, it presents suboptimal properties as high lipophilicity, which could result in low free fraction and poor solubility [17,33]. Additionally, the presence of the carboxylic moiety can lead to the formation of reactive metabolites such as acyl glucuronides, compounds chemically reactive leading to covalent binding with macromolecules and toxicity [49,50]. EVP-0015962 (**8**) move on to a phase II clinical trial in 2012 but results were not publicly reported. The latest update information in the platform *clinicaltrials.org* was in January 2014. Additionally, EVP- 0015962 (**8**) is not part of the AD pipeline in 2018, leading to the assumption that the clinical development of this compound was discontinued.

##### Non-NSAID Derived GSM’s

One of the first GSM series not presenting a carboxylic acid moiety (non-NSAID) was developed by Neurogenetics in 2004, leading to the discovery of NGP555 (Figure 11) [33].

This class of compounds share a scaffold consisting of four consecutives linked (hetero)aromatic rings identified as A, B, C and D which focus on aryl- or heteroarylimidazoles with an anilinothiazole. Figure 12 represents the basic scaffold and the structure-activity relationship established. The addition of a methyl or a halogen at the 4-position of the imidazole ring does not have a significant impact on potency, while the addition of a CF_3_ substituent in ring B, *ortho* to the thiazole, leads to a decrease in activity. A pyridine or pyrimidine ring at B ring increases potency, as well as the addition of an *ortho* methyl substituent in the aniline [51].

Compound NGP555 (**11**) from Neurogenetics (Figure 11) showed a decrease in CSF Aβ_42_ between 20–40% and an increase of the shorter forms in rodent studies. Additionally, it demonstrated protection from cognitive decline in two independent mouse studies using different memory and learning tasks [51]. NGP 555 (**11**) entered in phase I clinical trials in 2015. According to a press release from Neurogenetics on January 2017, NGP555 showed to be safe and well-tolerated in healthy volunteers [52]. Detailed results and future clinical studies with this compound have not been disclosed yet.

Based on this A-D scaffold, Eisai Pharmaceuticals developed a series of patented diarylcinnamide derivatives (**12**–**15**, Figure 13) [53], where the aminothiazole group present in Neurogenetics series was replaced by an α, β-unsaturated amide or a piperidone.

Beyond the common A-D scaffold, the molecules **12**–**15** developed shared an hydrogen bond donor (as a α, β-unsaturated amide or a piperidone) suggesting the importance of a hydrogen bond donor in this region [33]. The work around this cinnamide series from Eisai lead to the discovery of the clinical compounds E2012 (**16**) and E2212 (**17**) (Figure 14) [33].

E2012 (**16**) decreased levels of Aβ40 and Aβ42 in rat CSF, brain and plasma in vivo in a dose dependent manner. The reported IC_50_ values of E2012 (**16**) for Aβ40 and Aβ42 were 330 and 92 nM, respectively [54]. In rat CSF, E2012 (**16**) significantly decreased Aβ42 levels by 16.6% and 47.2% at doses of 10 and 30 mg/kg, respectively. It was also revealed that E2012 (**16**) reduced Aβ40 and Aβ42 and increased shorter Aβ peptides, such as Aβ37 and Aβ38, without changing total amount of Aβ peptides [55]. E2012 (**16**) was the first non-carboxylic acid to enter in clinical trials in 2006 and it showed to be efficacious in reduce plasma levels of Aβ42 of ~ 50% in a phase I clinical trial [56]. However, lenticular opacity was observed in a high-dose group of a 13-week preclinical safety study in rats, running in parallel to the phase I study leading to the suspension of the clinical study. Follow-up studies up to the highest dose tolerated in monkeys for E2012 (**16**) did not show ocular toxicity [57]. However, Eisai decided to develop their improved E2212 compound (**17**), instead [17].

E2212 (**17**) entered in a phase I clinical trial in 2010 (clinicaltrials.gov Identifier: NCT01221259). It demonstrated to have a similar pharmacological profile as E2012 (**16**) and a better safety profile, with no clinically significant ophthalmologic findings [58]. The PD response measured in plasma increased with the dose and was shown to perform 54% Aβ42 reduction at the 250 mg dose. No further development has been reported to date for E2212 (**17**) by Eisai and the platform *clinicaltrials.gov* does not present new studies. The predicted structure of E2212 (**17**), possessing four aromatic rings, a high molecular weight (480 g/mol) and a high cLogP (5.5) [17] could have conditioned its further development due to its poor drug-like properties. 

Other pharmaceutical companies, namely Schering, Roche, AstraZeneca, Bristol-Myers Squibb, Amgen, between others, had their own programs for the development of non-NSAID derived GSM [59]. The scaffold of the four consecutive linked rings firstly reported by Neurogenetics can be found in most of the non-NSAID compounds developed [33]. Consequently, they share the same suboptimal drug-like properties as the first non-NSAID derived GSM, leading to a lack of compounds reaching clinical evaluation. The lack of information about the structural characteristics of γ-secretase has been pointed as an important factor for the difficulty in developing potent GSM compounds [33]. The recently solved structure of gamma-secretase bound to the C99 fragment could help the development of potent and selective GSM compounds [60].

### 3.2. BACE-1 Inhibitors

BACE-1 is a type-1 membrane-anchored aspartyl protease responsible for the first step of the proteolysis of APP, identified in 1999 [61]. BACE-1 cleaves APP in the luminal surface of the plasma membrane and releases the soluble ectodomain of APP, leaving C99 (Aβ plus AICD) in the membrane to be subsequently cleaved by GS to generate Aβ peptides of different lengths as previously described (Figure 15) [61,62].

APP mutations close to the β-cleavage site that increase the efficiency of β-cleavage and result in overproduction of Aβ peptides strongly influence the risk of AD [52]. On the other and, a mutation adjacent to the β-cleavage site that reduces the formation of amyloidogenic peptides has a strong protective effect against AD [19]. Considering this genetic information, the inhibition of APP proteolysis by BACE-1 to lower the concentration of neurotoxic Aβ peptides became a rational strategy for clinical intervention.

BACE-1 protease is characterized by a large catalytic domain which is marked by the centrally located catalytic aspartates Asp32 and Asp228. Free BACE-1 features a flap-open conformation that is energetically stable due to the multiple hydrogen bonds in the flap region of the enzyme. When a substrate is bound, BACE-1 assumes a flap-closed or a flap-open conformation, depending on the characteristics of the substrate [16,63].

The catalytic domain of BACE-1 contain eight pockets consisted of different amino acid residues (Table 4).

#### 3.2.1. BACE-1 Inhibitor Development

Initially, the development of BACE-1 inhibitors appeared to be a relatively simple approach. First, the development of successful clinical aspartic proteases inhibitors for other therapeutic areas, namely human immunodeficiency virus (HIV) and hypertension, had established an important knowledge for the development of others aspartic protease inhibitors [20]. Second, the first crystal structure of this secretase, elucidated in 2000, provided powerful information for the structure-based drug design of BACE-1 inhibitors [65]. However, progress has been difficulted by combination of properties needed for being efficacious: compounds must fulfill the general rigid prerequisites for a drug aimed for CNS penetration and at the same time be compatible with the large and hydrophobic catalytic pocket of BACE-1. Moreover, selectivity toward different aspartyl proteases have been an additional attrition factor in BACE-1 drug discovery.

Nevertheless, despite the many challenges in the design of selective and effective BACE-1 inhibitors, several pharmaceutical industries have made impressive efforts for the improvement of BACE-1 inhibitors, with several compounds currently under clinical development.

Peptidomimetics, compounds mimicking the sequences of BACE-1 substrates were the first BACE-1 inhibitors developed and showed potent activity in vitro [15]. However, these large compounds suffer from poor PK properties, such as low bioavailability and low penetrance across the BBB, leading to the unsuccess of their development. Consequently, the design of BACE-1 inhibitors has focused on small molecules with nonpeptidic nature with improved PK properties and BBB penetration. Herein, a general overview of the structural evolution of BACE-1 inhibitors with a focus on the medicinal chemistry aspects of drug development programs will be provided.

##### Acyl-Guanidine-Based Inhibitors

A series of acyl guanidine-based BACE-1 inhibitors were discovered by high-throughput screening (HTS) by Wyeth (acquired by Pfizer in 2009) [63] represented by compounds **18**–**21** in Figure 16. An X-ray crystal structure of compound **18** complexed with the catalytic domain of BACE-1 revealed that the acyl guanidine moiety forms four key hydrogen interactions with the catalytic aspartic acids Asp32 and Asp228 [63]. A structural change in BACE-1 upon binding with the compound was observed, in which the flap region adopts an “open conformation”, due to stabilized interactions between Tyr71 and the π-system of the diarylpyrrole ligand **18** [63].

Additionally, it was observed that the two aryl groups extend into the S1 and S2′ pockets and the *para* position of the P_1_ phenyl group projects directly to an unoccupied S3 subsite, allowing the addition of substituents in the P_1_ phenyl and potentially increasing binding affinity. In contrast to the S1-S3 pockets, S2′ provides access to more polar/charged groups, allowing to explore analogues of compound **18** able to form hydrogen-bonds in this region [63].

According with the information substitutions were made to compound **18** in order to improve potency. Compound **19,** containing a *para* propyloxyphenyl moiety in the unsubstituted aryl ring and a propyl alcohol in the third guanidine nitrogen, allowed an improvement of approximately 30-fold in potency comparing with compound **18**. The 2-chloro group in P_2_ does not appear to contribute significantly to potency (Figure 17) [63].

Bristol-Myers Squibb (BMS) also worked in an acyl guanidine series (Figure 18). Starting with hit compound **22** it was developed compound **20** with a good potency against BACE-1 (IC_50_ = 5.0 nM) and inactive against other aspartyl proteases tested, namely cathepsin D (CatD), cathepsin E (CatE) and pepsin (IC_50_ > 100.000 nM) [66].

The optimized compound **20** was evaluated in rats in order to access its effect on Aβ40 levels in plasma, brain and CSF. Although a marked and dose-dependent reduction was observed in plasma Aβ40 (about 80%) no significant reduction in brain and in CSF was achieved (<20%). This lack of efficacy on brain and CSF was ascribed to P-glycoprotein (P-gp) efflux. Further improvement was necessary to maximize its PK properties [66].

Array BioPharma together with Genentech developed a series of chromane-based spirocyclic acyl guanidine-derived BACE-1 inhibitors leading to compound **21** (Figure 19). Although it showed a good selectivity to BACE-1 and was able to reduce CSF Aβ40 levels from 53% to 63% in rat and cyno, respectively, this compound also showed a high efflux ratio by P-gp [67].

##### 2-Aminopyridine-Based Inhibitors

A key advance in the development of small molecule BACE-1 inhibitors was the discovery of 2-amino heterocycles that, comparing with the initial acyl guanidine compounds, generally enable a better physico-chemical profile, improved brain penetration and in vivo efficacy [68].

Wyeth developed a series of pyrrolyl 2-aminopyridines, as an extension of their work on acyl guanidine inhibitors represented in Figure 16. Acyl guanidine inhibitors are polar compounds as suggested by the high total polar surface area (TPSA) ≈80 particularly due to the acyl guanidine moiety, which was associated to the poor BBB permeation (<5%) observed with this class of compounds. The bioisosteric replacement of the acyl guanidine moiety in compound **18** by an aminopyridine (compound **23**) was performed to improve compound permeability, maintaining the hydrogen-bonding interactions with the aspartic acids in the catalytic site of BACE-1 [69] (Figure 20).

The similarity of the hydrogen interaction between these two compounds was confirmed by X-ray studies which demonstrate the practically total sobreposition of the two moieties interacting with the aspartic acids Asp228 and Asp32 of the catalytic site of BACE-1 [69].

The modulation of the TPSA parameter enhanced the permeability, with compound **23** showing a good central drug exposure with a brain to plasma ratio of 1.7, compared with 0.04 achieved with the acyl guanidine-based compound **18** [69]. Further development of **23** by Wyeth led to compound **24** in Figure 21. 

The pyrimidine linked by an oxygen at *para* position of the P_1_ phenyl allow the establishment of hydrogen-bonding interactions with Ser229 at S3 region. The oxo-ethyl-hydroxyl at position-3 of the aminopyridine moiety allow an improved ligand affinity in S1′ region and selectivity against other proteases (Figure 22) [69].

The aromatic rings linked to the pyrrole ring established van der Waals interactions at the S1 and S2′ pockets, while the pyrrole ring of the ligand points toward the FLAP region, presumably making a π-edge stacking interaction with Tyr71 [69]. Compound **24** showed an IC_50_ (BACE-1) = 40 nM and >100-fold and >500 fold selectivity against BACE-2 and CatD, respectively.

##### Aminothiazine- and Aminooxazoline-Based Inhibitors

F. Hoffmann-La Roche (Roche) developed a series of aminothiazine-based inhibitors starting with the aminothiazine fragment **25** as a hit (Figure 23). X-ray analysis of hit 25 co-crystallized with BACE-1 showed that both nitrogens of the protonated amidine moiety form tight hydrogen bonds to the catalytic aspartates. Interactive three-dimensional (3D) modeling revealed that the S3 pocket of the catalytic site of BACE-1 could be best reached by meta-substitution on the phenyl ring. This meta-substitution was made with an amide linker to enable the formation of a hydrogen bond between the NH of the amide bond with the backbone carbonyl oxygen of Gly291 (Figure 24).

Additionally, it leads to a conformationally favorable fixation of the two aromatic rings in an almost planar arrangement. Extensive SAR studies led to the discovery of inhibitor 26, a highly active in vitro compound (BACE-1 enzyme IC_50_ = 27 nM, cellular assay IC_50_ = 2 nM). However, it turned out to be a good P- gp substrate (P-gp efflux ratio (P-gp ER) = 5.5), leading to unsatisfactory results in the in vivo model (<10% reduction of Aβ40 levels with an oral dose of 30 mg/kg) [26].

In order to improve brain penetration, Roche continued developing compound **26**. A set of modifications and SAR studies led to the discovery of an aminooxazine-based inhibitor containing a CF_3_ group. Compound **26** has a high basicity (pK_a_ = 9.8), giving rise to potential phospholipidosis and leading to the high efflux rate by P-gp. The introduction of fluorines into the headgroup showed to be efficacious in reducing pK_a_ about 3.5 log units and improving brain penetration. 

Compound **27** (Figure 25) was the most potent compound, combining favorable in vitro properties (cellular IC_50_ (BACE-1) = 2 nM, P-gp ER = 1.9) with a reduction of Aβ40 of 78% at a dose of 1 mg/kg in mice. With regards to its toxicological profile, compound **27** did not inhibit cytochromes P450 (CYP450) 3A4, 2D6, and 2C9 (IC_50_ > 25 μM) and showed to be selective against other aspartyl proteases such as human CatD /E and the peptidases renin and pepsin (IC_50_ > 200 μM) [70]. Amgen reported a series of small-molecule BACE-1 inhibitors including a xanthene core with a spirocyclic aminooxazoline head group. The company started with lead compound **28** (Figure 26) which showed a good Aβ lowering activity in a rat PD model, however it also has a low therapeutic window to QTc prolongation, consistent with in vitro activity on the human ether-a-go-go gene (hERG) ion channel [68,71].

Introduction of polar groups is used as a common strategy to reduce hERG binding affinity. However, it increases TPSA, which is strongly correlated with increased P-gp recognition. A series of SAR studies were conducted in order to determinate how to balance hERG, P-gp ER and BACE-1 potency by identifying regions of the molecule where polarity could be incorporated to minimize hERG activity without leading to a significant efflux or a potency decrease. This balance was accomplished by introducing polarity at the P2′ site and at the same time reducing the TPSA of the P3 group (Figure 27). The introduction of a fluorine in position 4 of the xanthene ring also improved BACE-1 enzymatic and cellular potencies, attributed to a favorable hydrogen-bond interaction between the 4-fluor atom and NH of Trp76 of BACE-1 [71].

Inhibitor **30** was orally administered to a number of species and showed a robust reduction of CNS Aβ40 levels (74 % and 75 % for CSF and brain, respectively) when orally administered in Dawley rats. It showed a bioavailability of 50% and 43% in rat and Cynomolgus monkey, respectively, and no significant effect on the QTc interval at a maximum dose of 12 mg/kg [72].

Among the several aminothiazine/aminooxazoline based inhibitors developed by different pharmaceutical research teams, just a limited number of compounds were able to reach clinical evaluation. Figure 28 depicted the structure of two aminothiazines (**31**–**32**) and one aminooxazine (**33**) based compounds with BACE-1 inhibitory activity that entered in clinical trials.

LY2811376 (**31**) is an aminothiazine based compound and was the first clinical BACE-1 inhibitor developed by Eli Lilly, which began phase I clinical trials in 2009. The inhibitor was given to 61 healthy individuals to investigate the safety and tolerability at single doses from 5 to 500 mg as oral capsules [73]. LY2811376 (**31**) showed to be well tolerated with no serious adverse effects reported. The maximum concentration was reached 2 h post dose in plasma and 5 h post dose in CSF, and a dose-dependent reduction of Aβ40 and Aβ42 was observed. With a single dose of 90 mg it was observed a Aβ40 reduction of 80% after 7 h and 54 % after 12–14 h, in plasma and CSF, respectively. However, in a rat toxicologic study performed in parallel with the phase I clinical studies, a retinal pathology at doses ≥ 30 mg/kg was observed, characterized by cytoplasmic accumulations of finely granular autofluorescent material dispersed within the retinal epithelium. As a consequence, the undergoing clinical trials of LY2811376 (**31**) were terminated and the compound did not proceed to phase II. This toxic effect was attributed to off-target effects of LY2811376 (**31**) against other aspartic acid proteases, namely CatD, as demonstrated by a subsequent study using LY2811376 (**31**) in BACE1 ^_/_^ mice [74]. As a safety procedure, all the study participants were contacted for follow-up exams 6–10 months after conclusion of the trial. Fortunately, all the volunteers revealed no clinically significant observations [74].

LY2886721 (**32**), also an aminothiazine based inhibitor, was the second clinical candidate tested in human subjects and the first BACE-1 inhibitor to reach phase II clinical trials. LY2886721 (**32**) was tested in six phase I clinical studies for the evaluation of its safety, tolerability, and pharmacology, evolving a total of 150 healthy volunteers. Fourteen days of daily dosing of a 70 mg strength reduced CSF Aβ40 by 74 % and CSF Aβ42 by 71 %. No safety concerns were apparent in dosing up to six weeks [75]. Based on the satisfactory results found in the phase I clinical trials, in March 2012 Lilly started a phase II study to examine the safety, tolerability, and PD effects of LY2886721 (**32**) in patients with mild AD [76]. During the study, routine safety monitoring detected abnormal liver enzyme elevations in 4 patients of 70. Consequently, Lilly voluntarily terminate the phase II study and the clinical development of LY2886721 (**32**) was halted. Again, the toxicity of this BACE-1 inhibitor was considered to be an off-target effect of the compound unrelated to BACE-1 inhibition [75,77].

RG7129 (RO5598887, **33**) is an aminooxazine-based compound developed by Roche which entered in clinical trials phase I in September 2011. Preclinical evaluation showed high potency against BACE-1 (IC_50_ = 30 nM). It also showed selectivity against CatD and CatE, pepsin and renin (>1000 fold) but not against BACE-2 (IC_50_ = 40 nM). Three phase I clinical trials have been completed between 2012 and 2013, however no results were reported. In October 2013, Roche terminated the development of RG7129 (**33**) but did not provide an explanation for its cessation [78]. In 2014, in a peer reviewed paper Roche reported a mouse study supporting the use of a combination treatment of BACE-1 inhibitor RG7129 (**33**) and the anti-Aβ antibody gantenerumab, suggesting a future clinical evaluation of these two compounds combined [79].

##### Aminoimidazole-Based Inhibitors

Starting with the aminoimidazole HTS hit **34** (Figure 29) Merck developed a series of aminoimidazole-based inhibitors represented in Figure 30 [16].

Molecular modeling studies showed that the amino group of the imidazole heterocycle was responsible for the binding with the catalytic aspartic acid residues of BACE-1. It was also observed that 2-methoxy-5-nitro substituents on the benzyl subunit led to potent inhibitors, such as compound **35** (Figure 30), however, their presence was also responsible for a high P-gp ER. The following synthesis of compound **36** allowed a significant reduction in P-gp ER and Merck moved forward to improve the potency of this inhibitor. The introduction of a conformational constraint in compound **37** allowed an increase in potency of 5 fold due to additional hydrophobic interactions with the flap region of BACE-1 [80]. Additionally, the introduction of a fluor group in compound **38** allowed additional hydrophobic interactions and increased the potency in about 6-fold to an IC_50_ (BACE-1) = 63 nM. This inhibitor has a reduced P-gp ER of 3.6, suggesting viable brain penetration [80].

Wyeth reported the design and synthesis of potent BACE-1 inhibitors with a bicyclic aminoimidazole scaffold, based on the HTS hit compound **39** (Figure 31). It represents a bicyclic aminoimidazole core and a biphenyl moiety, and a suboptimal potency (IC_50_ (BACE-1)) in micromolar range. X-ray studies show that the hit occupies the center of BACE-1 binding pocket (S1, S2′ regions) in an orientation where the aminoimidazole portion of the ligand directly interacts with the catalytic-site aspartic acids (Asp32 and Asp228) via hydrogen interactions. 

The structure also shows that the S3 region could be explored through the substitution of the moiety occupying the S1 pocket, indicating an opportunity to build off the phenyl moiety into this S3 region and improve binding affinity [81]. In order to achieve the unoccupied S3 region the *meta*-substitution of the phenyl ring was explored with benzyl analogues allowing an improvement of ligand potency about 5-fold. Greater results were obtained with a pyridine moiety where the pyridine nitrogen interacts with Ser229 through the conserved water at S3 pocket (Figure 32). The introduction of a 2-fluorine was centered on subtle metabolic properties (additional information not disclosed). In additional SAR studies exploring the phenyl ring that projects into the S2′pocket, it was discovered that the trifluoromethoxy substituted compound **40** shows an approximate 15-fold improvement of BACE-1 potency (comparing with compound **39**). The improved compound **40** (*R*-enantiomer) showed a high potency for BACE-1 inhibition and >100-fold selectivity over the other structurally related aspartyl proteases BACE-2, CatD, renin, and pepsin [81].

Independently from Wyeth, AstraZeneca also developed a series of inhibitors based on compound **39**, herein represented by compounds **41** and **42** in Figure 33.

Compound **41**, with a *p*-difluoromethyl ether substitution on one of the phenyl rings and an *m*-alkynyl substituent on the other, showed good potency in both enzymatic and cellular assays with pIC_50_ values of 7.11 and 7.46, respectively. Cell membrane permeability, as determined by a Caco-2 assay, was 8.4 x 10^-6^ cm/s, and the efflux ratio was 3.5, indicating potential for BBB penetration. The crystal structure of compound **41** in complex with BACE-1 displayed the interaction of the aminoimidazole moiety with the catalytic Asp32 and Asp228 residues, binding in a flap-open conformation, allowing Trp76 to be in position for hydrogen bonding to the oxygen of the *p*-difluoromethyl ether. The alkynyl substituent of the second ring extends into the S3 pocket [16,82].

Replacement of the alkyne chain of inhibitor **41** with a fluorinated propyl ether resulted in inhibitor **42**. This inhibitor showed similar potency as compound **41** but an enhanced efflux ratio of 0.8, which is 4 times lower. However, an in vivo assessment in a mouse model using oral administration of compound **42** showed a low brain/plasma ratio of 0.18, showing that the results obtained in the Caco-2 cells efflux test were underestimated. This lack of brain penetration was attributed to P-gp efflux, once the coadministration of **42** with a P-gp inhibitor resulted in a brain/plasma ratio of 2.34 [82].

Compound AZD3293 (LY3314814, Lanabecestat, **43**, Figure 34) is an aminoimidazole based compound developed by AstraZeneca which reached phase I clinical trials in 2012 [83].

The results from the first two phase I clinical studies are currently available [84]. It was composed of (1) a single ascending dose study evaluating doses of 1–750 mg with a food-effect component (*n* = 72), and (2) a 2-week multiple ascending dose study evaluating doses of 15 or 50 mg once daily or 70 mg once weekly in elderly subjects (Part 1, *n* = 31), and 15, 50, or 150 mg once daily in patients with mild to moderate AD (Part 2, *n* = 16). Results showed that AZD3293 (**43**) was generally well tolerated up to the highest doses given. No notable food effects were observed. For single doses ≥5 mg, ≥70% reduction was observed in mean plasma Aβ40 and Aβ42 concentrations, with prolonged suppression for up to 3 weeks at the highest dose level studied. Following multiple doses, robust reductions in plasma (≥64% at 15 mg and ≥78% at ≥50 mg) and CSF (≥51% at 15 mg and ≥76% at ≥50 mg) Aβ peptides were seen, including prolonged suppression even with a once weekly dosing regimen [84].

In 2015 and 2016, four additional phase I trials with a total of 175 healthy volunteers were conducted. They evaluated a new tablet formulation and possible interactions of this inhibitor with several drugs commonly prescribed in the elderly, namely warfarin and dabigatran, midazolam, as well as simvastatin and donepezil [85]. Currently, AZD3293 (**43**) is being evaluated in two phase III clinical trials (NCT02245737 and NCT02783573) sponsored by an alliance between AstraZeneca and Eli Lilly [83]. These multi-center, randomized, double-blind, placebo-controlled studies are testing the disease-modifying potential of AZD3293 (**43**) at daily doses 20 or 50 mg for 18 to 24 months, in over 4,000 patients with mild cognitive impairment due to AD and mild AD [84].

##### Iminothiadiazinane Dioxide-Based Inhibitors

Merck developed a series of iminothiadiazinane dioxide based inhibitors represented by compound **45**, verubecestat. The starting point was an iminopyrimidinone scaffold (**44**) which was modified in order to improve binding affinity and explore a new intellectual property space [86] (Figure 35).

Although the PD activities of compound **44** were considered favorable for further clinical development, this compound did not advance. Metabolite profiling following oral administration to rats revealed biliary excretion of metabolites corresponding to direct glutathione addition and glutathione adducts derived from oxidative metabolism. Additionally, the modest CatD selectivity (only 21 fold) was considered to be inadequate and a major limitation to its progression [87]. In this way, an effort was made in order to remove the metabolically labile propynylpyridine and improve CatD selectivity.

Merck had previously disclosed the iminohydantoin **46** (Figure 36) which although represents a suboptimal activity, it displayed an enhanced selectivity for BACE-1 over CatD.

In an X-ray cocrystal structure of compound **46** with BACE-1, the interaction of the phenyl and furanyl groups with the S1 and S3 pockets, respectively, and the amide N-H engaged in a hydrogen-bonding interaction with the carbonyl of Gly230 were observed. Additionally, comparing the interactions of the iminopyrimidinone based compound **44** and the iminohydantoin **46** with BACE-1, it was observed that the furanyl ring of the iminohydantoin is projected deeper into the S3 subpocket (S3^sp^) of BACE-1 in comparison with the pyridyl ring of the iminopyrimidinone based compound **44** [87].

Comparing the crystal structures of BACE-1 and CatD it was observed that their S3^SP^ differ in topology, where the CatD S3^SP^ in slightly smaller than the one of BACE-1. Hence, it was postulated the differences in CatD selectivity of these two compounds is related with their interaction with the S3 domains of the two proteases [87].

In this way, Merck conducted a series of SAR studies in order to improve selectivity over CatD while maintaining high BACE-1 affinity, leading to the discovery of verubecestat (**45**, Figure 37). The pyridyl motif in verubecestat (**45**) allowed an improved selectivity over CatD >10 000. The 6-fluoro substituent in the phenyl group allowed an improvement of 5-fold in potency comparing with the defluoro analogues and the iminothiadiazinane dioxide core enhanced permeability and reduced efflux ratio, improving in vivo potency [87].

Verubecestat (**45**) showed good PK/PD properties in preclinical species, allowing a profound and sustained reduction of CSF Aβ40 levels in cynomolgus monkeys (72% and 81% reduction at 3 and 10 mg/kg, respectively). In a single-dose cardiovascular study in telemetered monkeys it showed no effect on the QT interval and no induction of CYP 3A4 or 1A2 expression in human hepatocytes [87].

Despite the fact verubecestat (**45**) has selectivity over the proteases CatD, CatE, pepsin and renin (>45,000, >45,000, >45,000, and 15,000, respectively) it showed to be a potent inhibitor of BACE-2 (K_i_ (BACE-2) = 0.38 nM against K_i_ (BACE-1) = 2.2 nM). Although the specific functions of BACE-2 are currently not well known, recent reports associate BACE-2 with the process of pigmentation, consistent with the lighter coat color observed in BACE-2 knockout mice [88]. This phenomenon was observed in mice and rats treated with verubecestat (**45**), but not in the chronic toxicology studies performed in monkeys. Overall, the relatively benign phenotypes of BACE-2 knockout mice, current understanding of the role of BACE-2 processing of its endogenous substrates, and the outcome of preclinical toxicology studies has mitigated concerns related to the lack of selectivity of verubecestat (**45**) over BACE-2 [87].

Verubecestat (**45**) advanced to phase I clinical trials in 2011 to assess safety, PK and PD in healthy volunteers and in mild to moderate AD patients. Single and multiple doses were generally well tolerated and produced reductions in Aβ levels in the CSF > 80% of both healthy human subjects and AD patients. Notably, there were no reports of changes in skin or hair pigmentation as a potential consequence of BACE-2 inhibition in any of the phase I studies; however, longer treatment time would likely be required for pigmentation changes to manifest [87]. In 2012, verubecestat (**45**) entered into phase II/III clinical trials in people with mild to moderate AD (EPOCH trial), and in 2013 a phase III trial for prodromal AD (APECS trial) was also started [68].

EPOCH trial was discontinued on February 2017 due to lack of efficacy, with researchers defending that it is very unlikely to observe a clinical benefit in using a BACE-1 inhibitor in cases that a substantial synaptic and neuron loss is already installed [89].

In February 2018, APECS clinical trial was also discontinued and Merck no longer listed verubecestat in its research pipeline. Participants taking verubecestat scored worse than the placebo group on the cognitive test Alzheimer’s Disease Cooperative Study-Activities of Daily Living (ADCS-ADL), in a small but significant way. Further research in needed to understand the origin of this negative effect, if it is reversible and if can be associated with certain patient populations or stages of disease [90,91].

## 4. Conclusions

The discovery of disease modifying agents for AD has shown to be very challenging for medicinal chemists. Compounds should exhibit improved CNS penetration by enhanced BBB permeability and reduced P-gp ER. The physical chemistry characteristics needed for this PK properties need to be balanced with target requirements, in which the improvements of these properties sometimes lead to a reduction on binding affinity and potency. Additionally, off-target effects also need to be addressed, being these features one of the principal causes for the discontinuation of clinical programs.

Medicinal chemistry teams have utilized high throughput and virtual screening followed by chemical optimization trying to accommodate several physico-chemical nuances and discover new compounds able to be safety and efficacy in modifying AD progression.

Taking into consideration the actual AD pipeline, a special focus has been given to BACE-1 as a target for a disease modifying therapy for AD. The information currently available pointed the inhibition of BACE-1 as a safety and efficacy target for Aβ reduction. All the adverse effects identified up to date in the use of BACE-1 inhibitors were pointed as off-target effects, confirming BACE-1 as a suitable target to explore. Nevertheless, medicinal chemists should perform extended SAR studies, not only for potency and PK properties exploration, but also in terms of selectivity, providing the discovery of compounds highly selective for BACE-1 against other related proteases such as CatD.

Another question stands on the adequacy of the pharmacological therapy with the stage of Alzheimer disease (AD). The use of disease modifying therapies as BACE-1 inhibitors should be a suitable option for early stages of AD where minimum neuronal loss and synaptic dysfunction are observed. On the other hand, in a mild-to-moderate AD scenario it would possibly be too late in the disease process for BACE-1 inhibition to be effective and a symptomatically therapeutic should be preferred.

In sum, the future of AD will rely on the development of potent, selective and safety compounds able to delay AD progression and neuronal impairment. At the same time, it is essential the identification of specific biomarkers to allow an early and efficacious pharmacological intervention.

## Figures and Tables

**Figure 1 pharmaceuticals-12-00041-f001:**
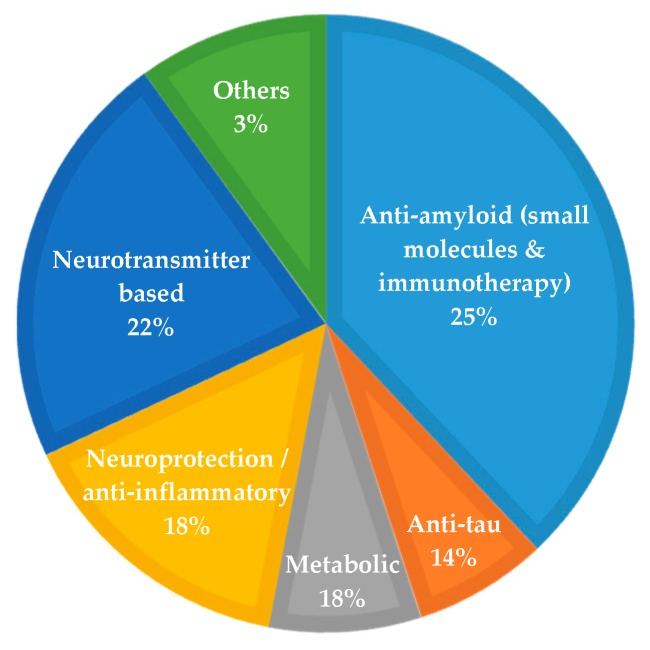
Mechanism of action of AD agents in clinical trials in 2018. Adapted from [14].

**Figure 2 pharmaceuticals-12-00041-f002:**
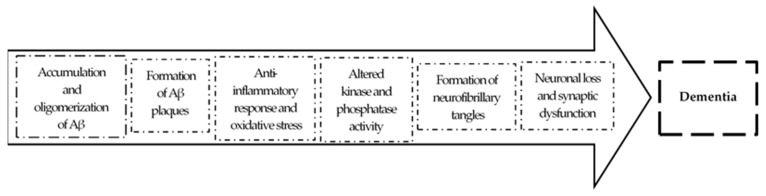
Cascade of events according the amyloid hypothesis. Adapted from [16].

**Figure 3 pharmaceuticals-12-00041-f003:**
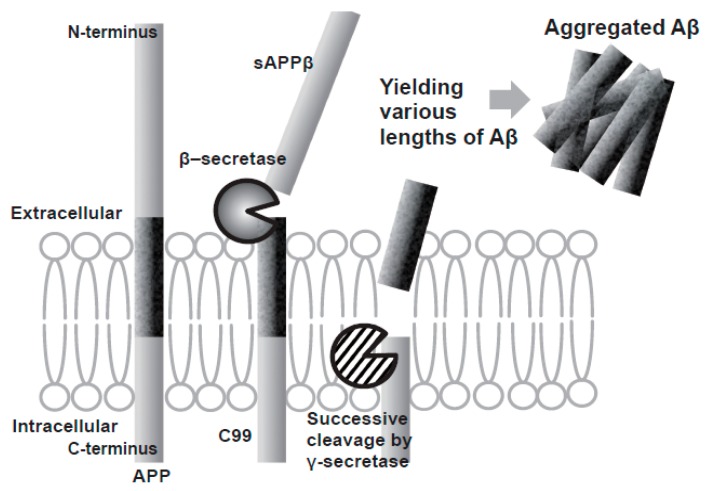
Scheme of the production of Aβ by the two step sequential cleavage of APP by β-secretase and γ-secretase. Adapted from [15].

**Figure 4 pharmaceuticals-12-00041-f004:**
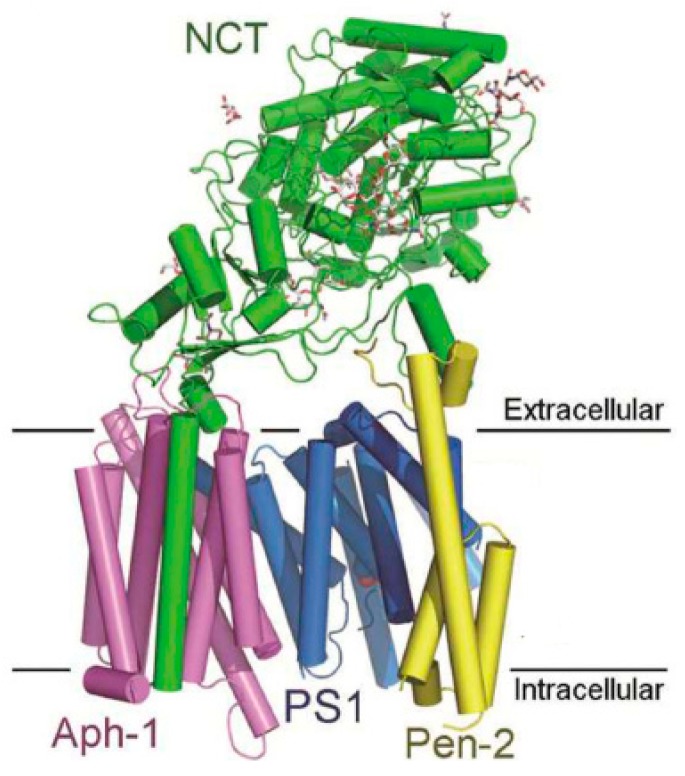
Crystal structure of human γ-secretase. Adapted by permission from Xiao-chen Bai et al. Nature [30], Copyright 2015.

**Figure 5 pharmaceuticals-12-00041-f005:**
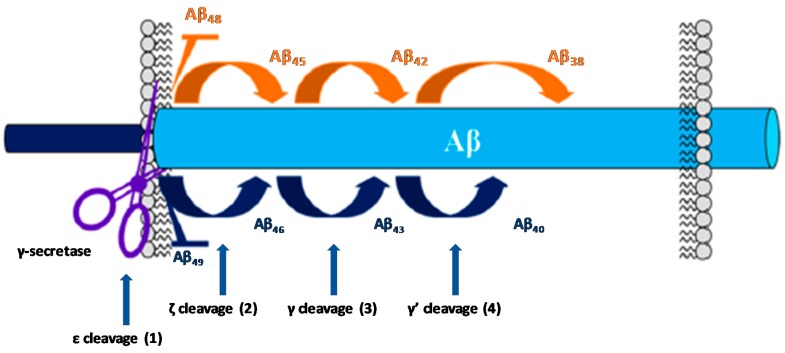
Cleavage steps of C99 by GS. Adapted with permission from [17]. Copyright 2016 American Chemical Society.

**Figure 6 pharmaceuticals-12-00041-f006:**
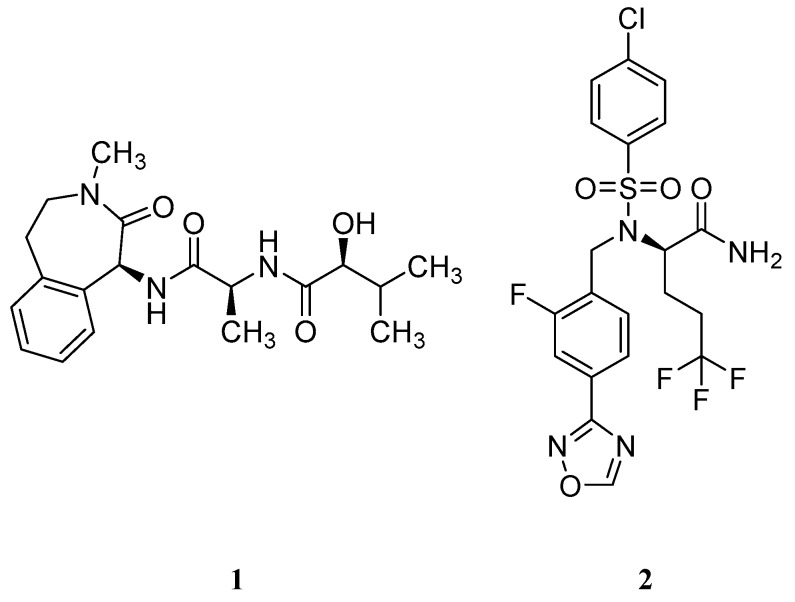
γ-Secretase inhibitors (**1**–**2**) advanced in late-stage clinical trials.

**Figure 7 pharmaceuticals-12-00041-f007:**
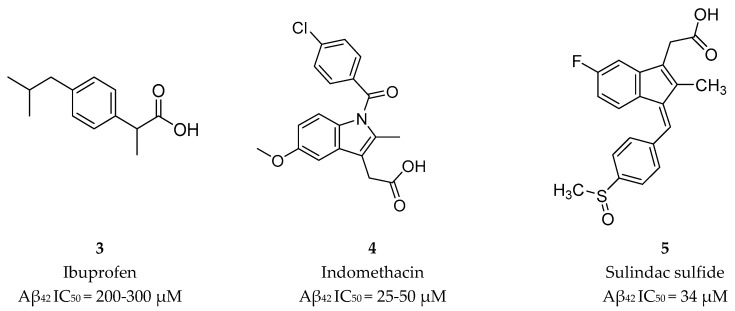
NSAIDs GSMs **3**–**5** and respective Aβ42 half-maximal inhibitory concentration (IC_50_) values [44].

**Figure 8 pharmaceuticals-12-00041-f008:**
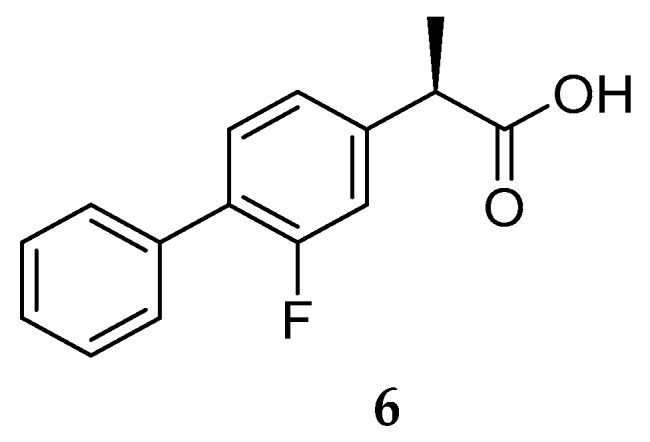
Tarenflurbil (R-flurbiprofen) (**6**).

**Figure 9 pharmaceuticals-12-00041-f009:**
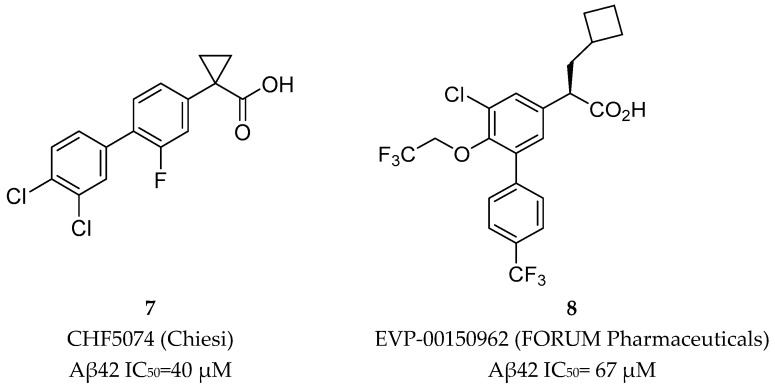
CHF5074 (**7**) and EVP-0015962 (**8**).

**Figure 10 pharmaceuticals-12-00041-f010:**
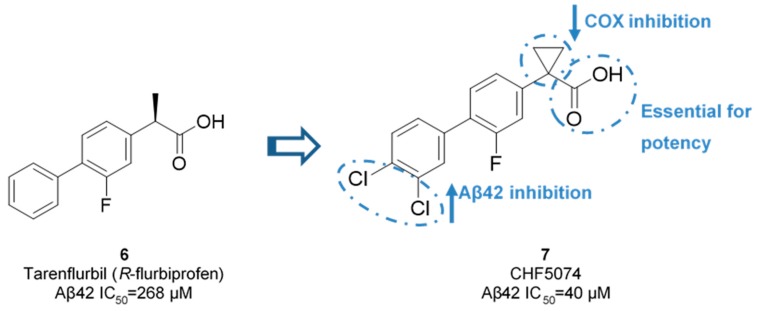
Improvement activity and selectivity of CHF5074 (**7**) against tarenflurbil (**6**).

**Figure 11 pharmaceuticals-12-00041-f011:**
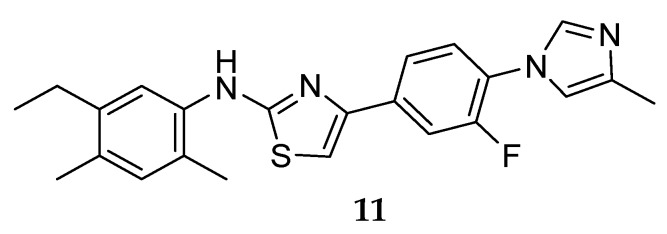
Structure of NGP555 (**11**).

**Figure 12 pharmaceuticals-12-00041-f012:**
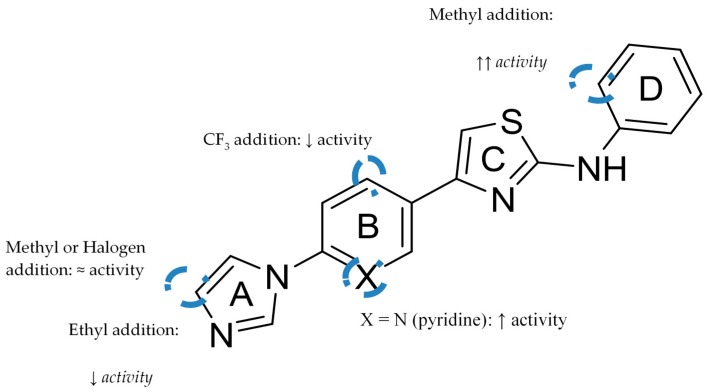
Structure-activity relationship (SAR) of non-NSAID GSMs four rings scaffold.

**Figure 13 pharmaceuticals-12-00041-f013:**
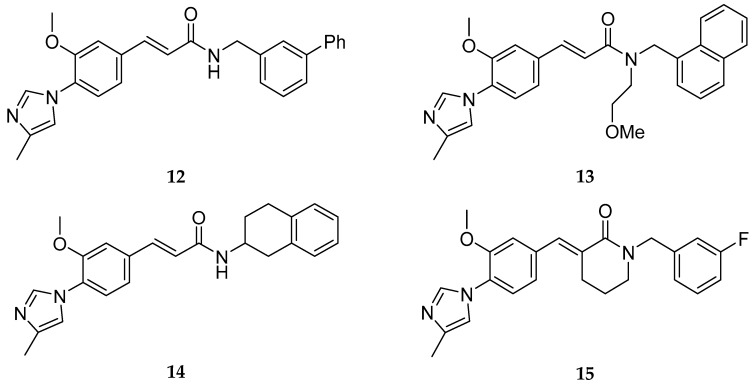
Examples of Eisai cinnamides.

**Figure 14 pharmaceuticals-12-00041-f014:**
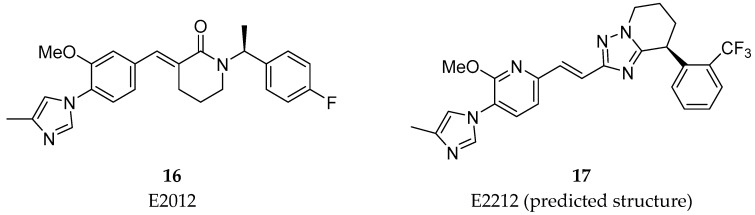
Structures of E2012 (**16**) and E2212 (**17**, predicted structure).

**Figure 15 pharmaceuticals-12-00041-f015:**
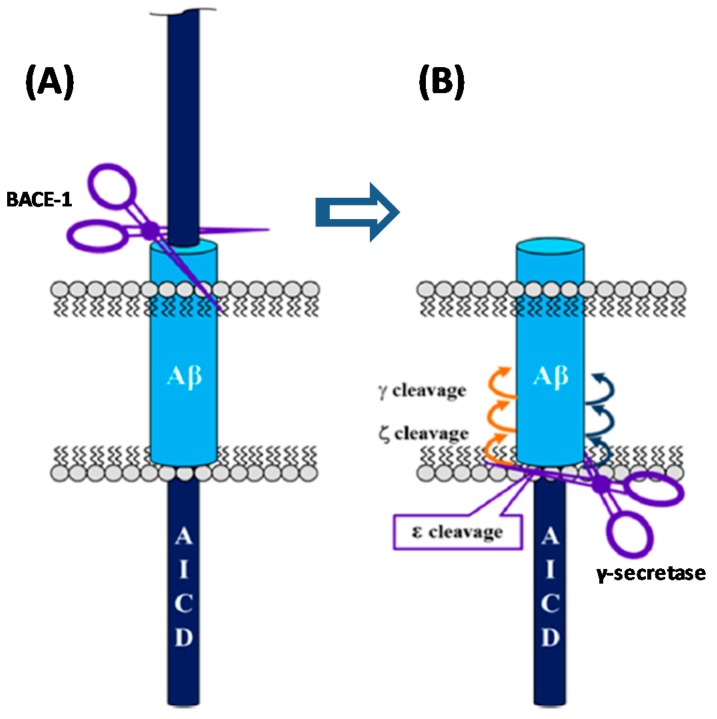
Processing steps of APP by (**A**) BACE-1 and (**B**) GS. Adapted with permission from [17]. Copyright 2016 American Chemical Society.

**Figure 16 pharmaceuticals-12-00041-f016:**
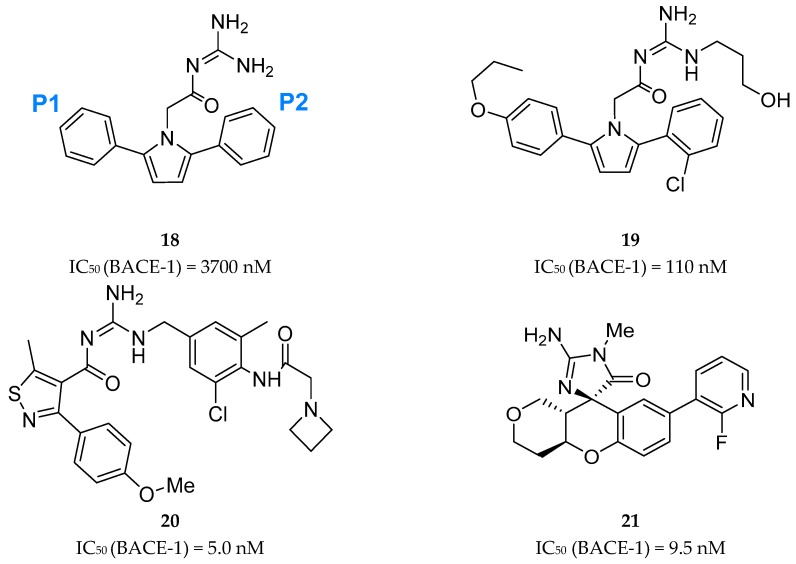
Structures and activity of acyl guanidine-based BACE-1 inhibitors **18–21**.

**Figure 17 pharmaceuticals-12-00041-f017:**
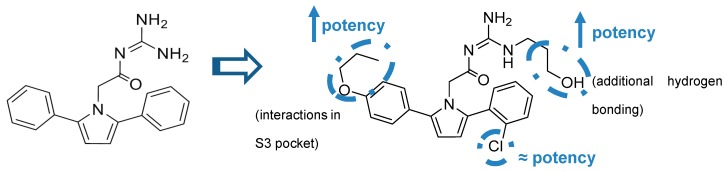
Optimization of **18** to **19**.

**Figure 18 pharmaceuticals-12-00041-f018:**
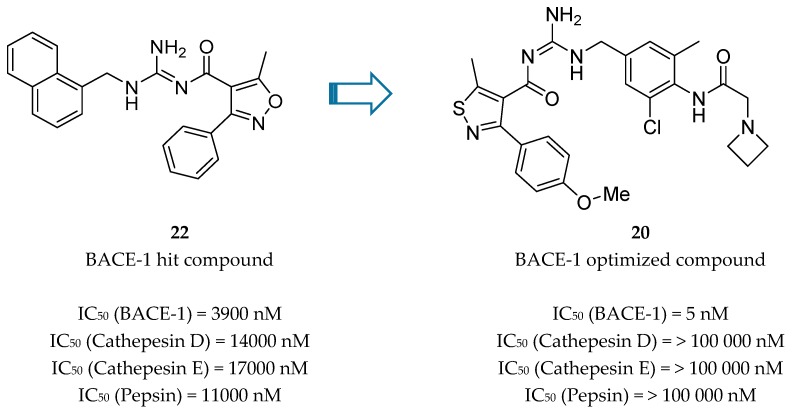
Acyl guanidine-based BACE-1 inhibitors developed by BMS [66].

**Figure 19 pharmaceuticals-12-00041-f019:**
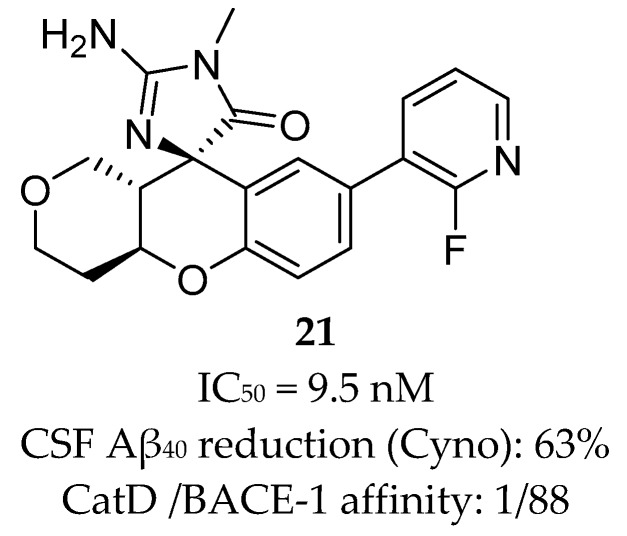
Chromane-based spirocyclic acyl guanidine-derived BACE-1 inhibitor **21** develop by Array BioPharma together with Genentech [67].

**Figure 20 pharmaceuticals-12-00041-f020:**
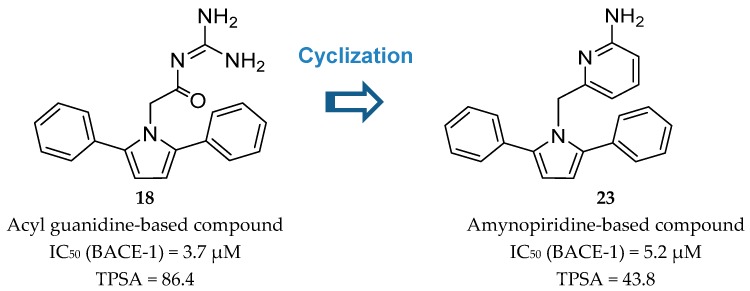
Bioisosteric replacement of the acyl guanidine moiety of compound **18**.

**Figure 21 pharmaceuticals-12-00041-f021:**
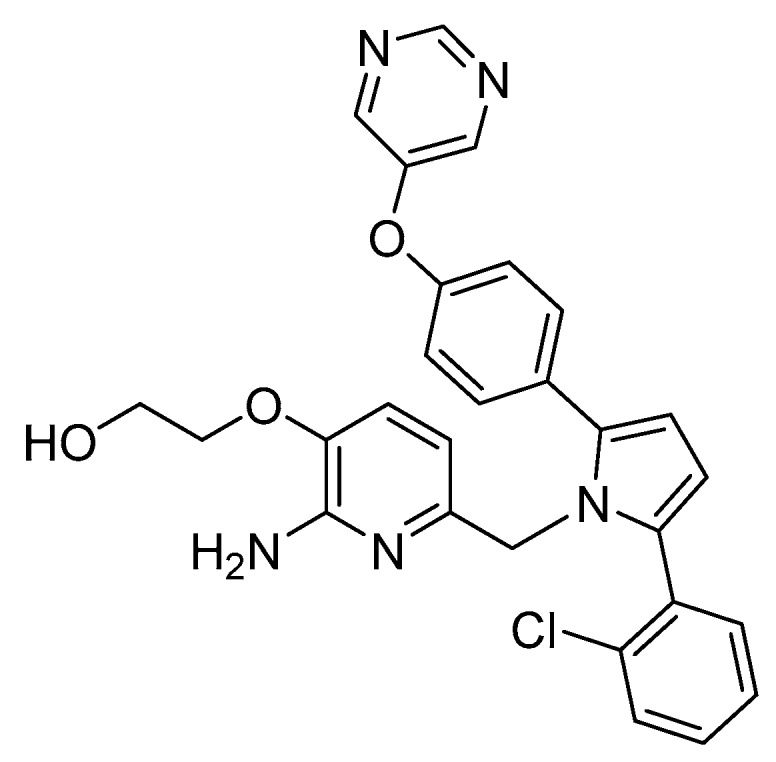
Structure of the improved aminopyridine-base compound **24** developed by Wyeth.

**Figure 22 pharmaceuticals-12-00041-f022:**
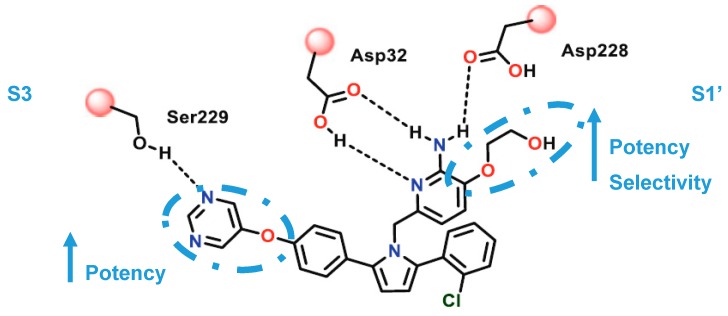
Interactions of the improved aminopyridine-base compound **24** developed by Wyeth with the catalytic site of BACE-1. Adapted from [16].

**Figure 23 pharmaceuticals-12-00041-f023:**
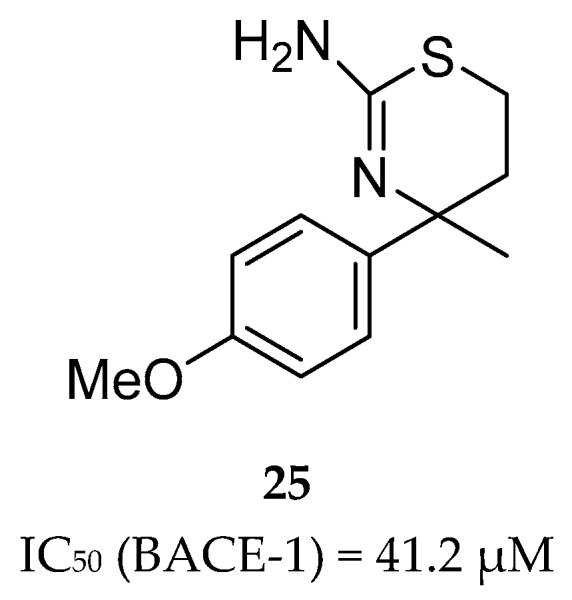
HTS aminothiazine fragment hit **25**.

**Figure 24 pharmaceuticals-12-00041-f024:**
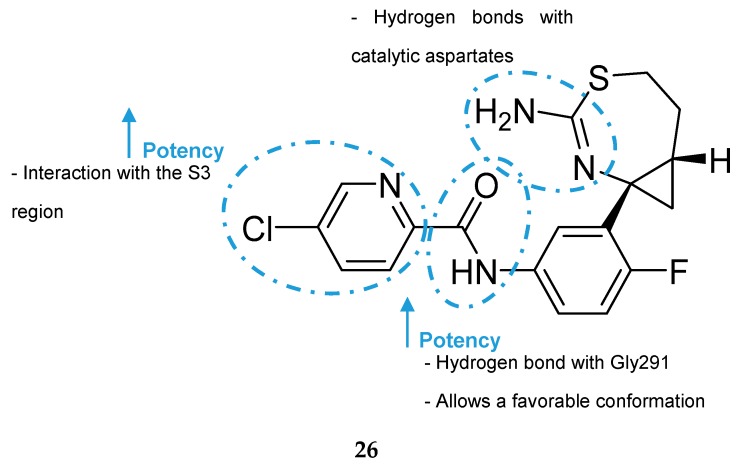
SAR of optimized inhibitor **26** developed by Roche.

**Figure 25 pharmaceuticals-12-00041-f025:**
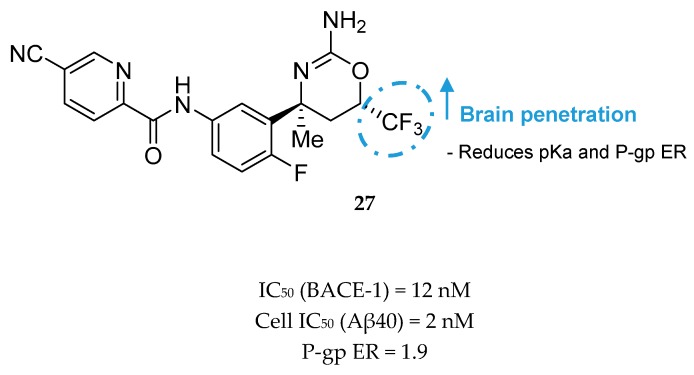
Oxazine-based compound **27** with a trifluoromethyl group developed by Roche.

**Figure 26 pharmaceuticals-12-00041-f026:**
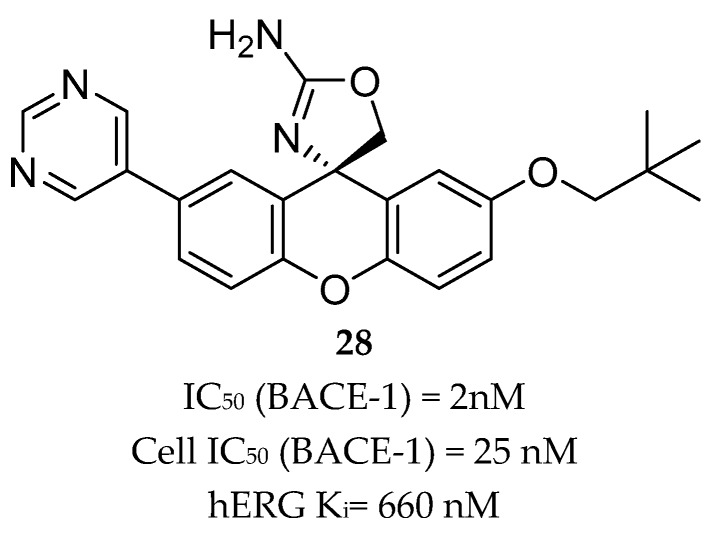
Spirocyclic aminooxazoline lead compound **28**.

**Figure 27 pharmaceuticals-12-00041-f027:**
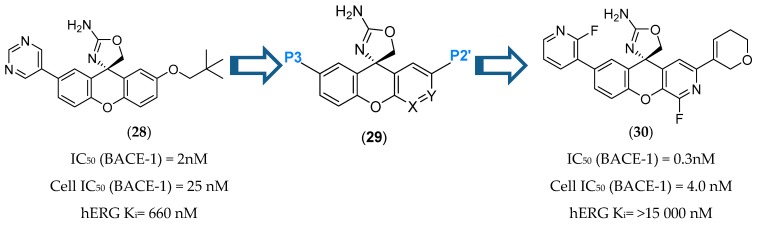
Spirocyclic aminooxazoline developed by Amgen [68,71].

**Figure 28 pharmaceuticals-12-00041-f028:**
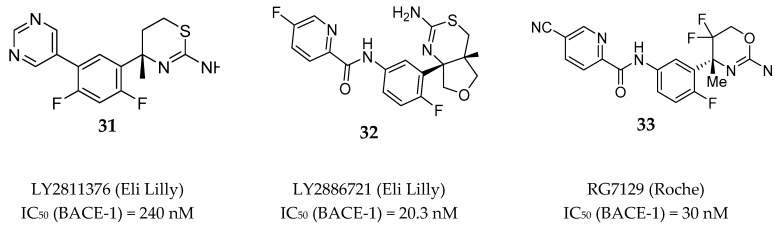
Aminothiazines **31–32** and aminooxazine **33** based compounds evaluated in clinical trials.

**Figure 29 pharmaceuticals-12-00041-f029:**
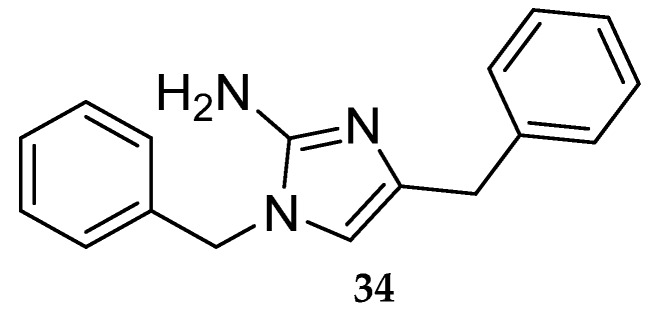
Merck’s aminoimidazole HTS hit.

**Figure 30 pharmaceuticals-12-00041-f030:**
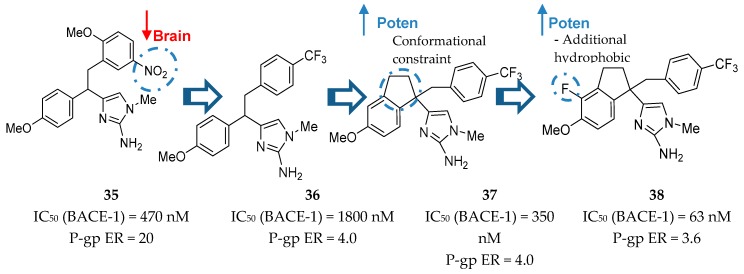
Merck’s aminoimidazole-based inhibitors.

**Figure 31 pharmaceuticals-12-00041-f031:**
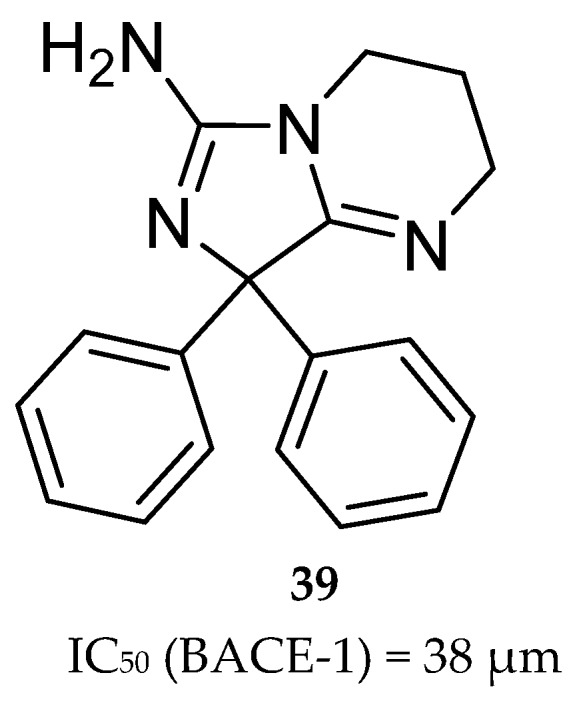
Bicyclic aminoimidazole hit compound **39**.

**Figure 32 pharmaceuticals-12-00041-f032:**
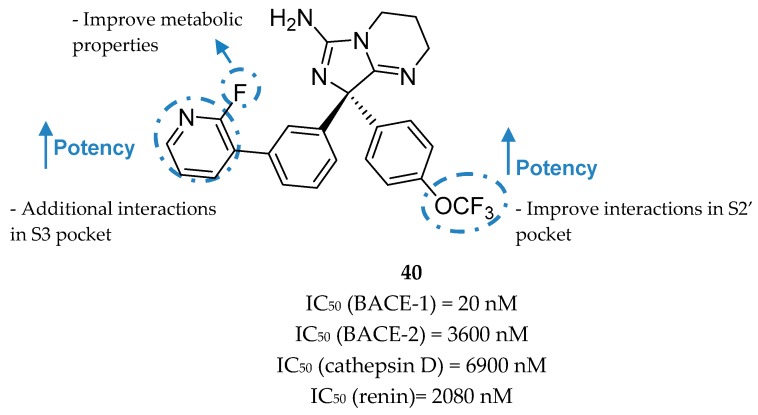
Optimized aminoimidazole-based inhibitor **40** from Wyeth.

**Figure 33 pharmaceuticals-12-00041-f033:**
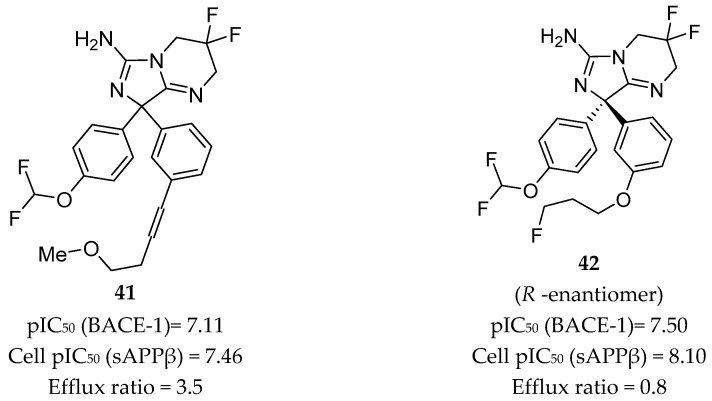
Aminoimidazole-based inhibitors **41**–**42** developed by AstraZeneca.

**Figure 34 pharmaceuticals-12-00041-f034:**
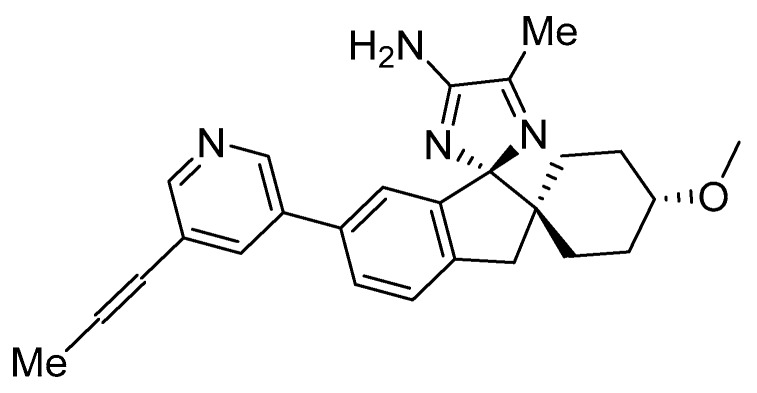
AZD3293 (LY3314814, lanabecestat, **43**) from AstraZeneca.

**Figure 35 pharmaceuticals-12-00041-f035:**
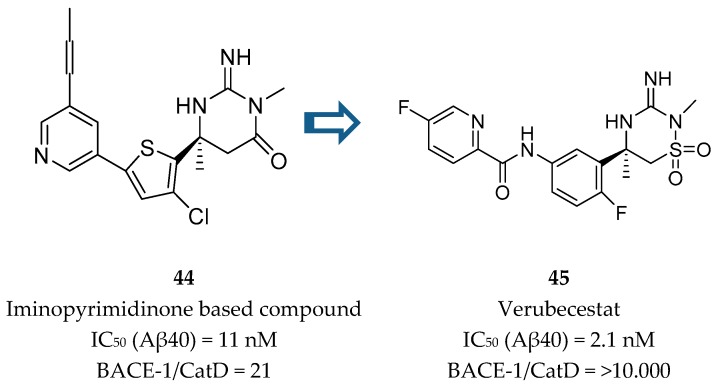
Structures of iminopyrimidinone based compound **44** and verubecestat (**45**).

**Figure 36 pharmaceuticals-12-00041-f036:**
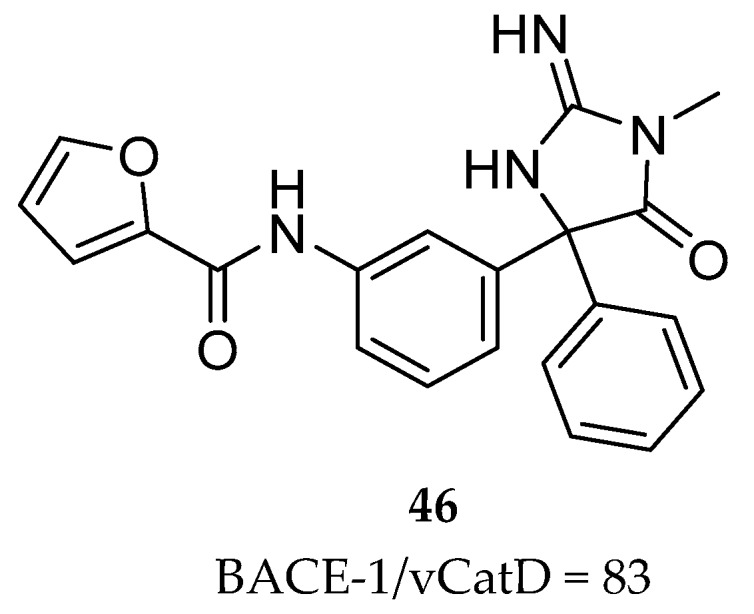
Iminohydantoin **46** previously developed by Merck.

**Figure 37 pharmaceuticals-12-00041-f037:**
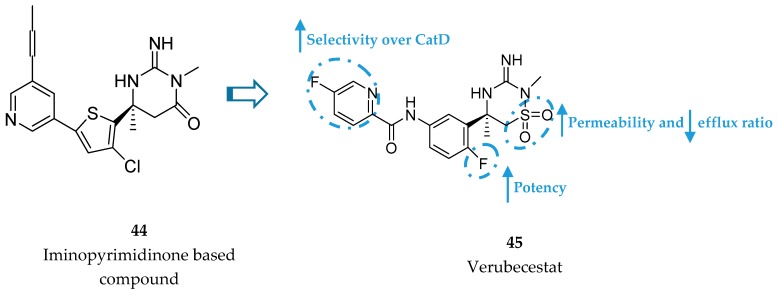
Drug development and SAR of verubecestat (**45**).

**Table 1 pharmaceuticals-12-00041-t001:** Approved drugs for the treatment of Alzheimer’s disease.

Name	Structure
Tacrine	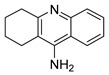
Donepezil	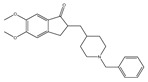
Rivastigmine	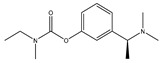
Galantamine	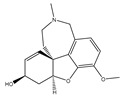
Memantine	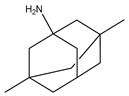

**Table 2 pharmaceuticals-12-00041-t002:** Immunotherapeutic agents (amyloid related) in the AD pipeline (2018).

Phase (Clinical Trial(s))	Agent	Mechanism of Action	Sponsor
I, III	Aducanumab	Monoclonal antibody	Biogen
III	Albumin + immunoglobulin	Polyclonal antibody	Grifols
I, II, III	Crenezumab	Monoclonal antibody	Roche/Genentech
II, III	Gantenerumab	Monoclonal antibody	Roche
III	Solanezumab	Monoclonal antibody	Washington University, Eli Lilly, Roche, NIA, Alzheimer’s Association, ATRI (Alzheimer’s Therapeutic Research Institute)
II, III	CAD106	Amyloid vaccine	Novartis, Amgen, NIA,
II	Sargramostim (GM-CSF)	Immunostimulator	University of Colorado, Denver, The Dana Foundation
II	BAN2401	Monoclonal antibody	Eisai
II	UB-311	Monoclonal antibody	United Neuroscience
I, II	LY3002813	Monoclonal antibody	Eli Lilly and Company
I	LY3303560	Monoclonal antibody	Eli Lilly and Company
I	Lu AF20513	Polyclonal antibody	H. Lundbeck A/S
I	KHK6640	Monoclonal antibody	Kyowa Hakko Kirin Pharma

**Table 3 pharmaceuticals-12-00041-t003:** Small molecules (amyloid related) in AD pipeline (2018).

Phase Clinical Trial(s)	Agent	Mechanism of Action	Sponsor
I	NGP 555	GSM	NeuroGenetic Pharmaceuticals
II	ID1201	Phosphatidylinositol 3-kinase/Akt pathway activation	II Dong Pharmaceutical Co
II	Nilotinib	Tyrosine kinase inhibitor	Georgetown University
III	CNP520	(γ-secretase modulator)	Alzheimer’s Association
III	ALZT-OP1a (cromolyn)+ ALZT-OP1b (ibuprofen)	BACE1 inhibitor	AZTherapies
III	Sodium Oligo-mannurarate (GV-971)	Increases amyloid clearance	Shanghai Green Valley
III	TTP488 (Azeliragon)	RAGE^1)^ (Receptor for advanced glycation end products) antagonist	vTv Therapeutics
II, III	JNJ-54861911	BACE1 inhibitor	Janssen
II, III	E2609 (Elenbecestat)	BACE1 inhibitor	Eisai, Biogen
II	LY3202626	BACE1 inhibitor	Eli Lilly
II	Atomoxetine	Adrenergic uptake inhibitor, SNRI	Emory University, NIA
II	AZD0530 (Saracatinib)	Kinase inhibitor	Yale University, ATRI,
II	CT1812	Sigma-2 receptor competitive inhibitor2)	Cognition Therapeutics
II	Posiphen	Selective inhibitor of APP production	QR Pharma, ADCS
II	Valacyclovir	Antiviral agent 4)	Umea University
III	AZD3293 (LY3314814)	BACE1 inhibitor	AstraZeneca, Eli Lilly

**Table 4 pharmaceuticals-12-00041-t004:** Pockets and their amino acid residues on the catalytic domain of BACE-1 [64].

Pocket	Amino Acid Residues	Pocket	Amino Acid Residues
S1	Leu30, Asp32, Tyr71, Leu119, Gln73, Phe108	S1′	Val31, Tyr71, Thr72, Asp228
S2	Asn233, Arg235, Ser325	S2′	Ser35, Val69, Pro70, Tyr198
S3	Leu133, Ile110, Ser229	S3′	Arg128, Tyr198
S4	Gln73, Thr232, Arg307	S4′	Glu125, Ile126, Tyr197, Tyr198

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
