# Peer review of "BACE-1 and γ-Secretase as Therapeutic Targets for Alzheimer’s Disease"

_pharmaceuticals, 2019, doi:10.3390/ph12010041_

Round 1
Reviewer 1 Report
In this review, the authors summarized the efforts of the pharmaceutical industry to develop therapeutics for Alzheimer's disease. The manuscript gives a thorough historical overview of the challenges faced by the field, and is in my opinion suitable for publication after the following issues have been addressed:
1. Despite the general title ('amyloid targeting strategies'), the review is mostly focussed on small molecule approaches, and inhibitors of gamma-secretase and BACE1. In the light of the various antibody based therapies under development, which are not described in detail, I therefore suggest to adapt the title to more appropriately reflect the content of the review.
2. In general, the manuscript is well written, but suffers from occasional spelling and grammatical errors, which should be corrected.
3. Chemical structures for galantamine and memantine are missing in Table 1.
4. Tables 2 and 3 seem to contain data from 2017, they should be updated to reflect notable failures in clinical trials from 2018 (i.e. Verubecestat).
Author Response
In this review, the authors summarized the efforts of the pharmaceutical industry to develop therapeutics for Alzheimer's disease. The manuscript gives a thorough historical overview of the challenges faced by the field, and is in my opinion suitable for publication after the following issues have been addressed:
1. Despite the general title ('amyloid targeting strategies'), the review is mostly focussed on small molecule approaches, and inhibitors of gamma-secretase and BACE1. In the light of the various antibody based therapies under development, which are not described in detail, I
therefore suggest to adapt the title to more appropriately reflect the content of the review.
We agree and changed the title to “BACE-1 and γ-secretase as therapeutic targets for Alzheimer’s Disease”. Moreover, to clarify this aspect the following sentence in the introduction section was modified: “With regard to small molecules, different targets have been identified for activity modulation hoping to get a marked impact in the amyloid cascade and in disease progression. Among 16 anti-amyloid agents identified in the AD pipeline, six have their mechanism of action centered in the activity of β or γ-secretase proteases, covering almost 40% of the identified agents. In fact, in the last years several pharmaceutical industries and other research centers have led research programs to discovered potent compounds for the modulation or inhibition of these two targets [16], [17], [24]–[26]. The discovery and optimization activities which lead to the development of these compounds will be herein detailed.”
2. In general, the manuscript is well written, but suffers from occasional spelling and grammatical errors, which should be corrected.
We thank the reviewer and checked in the revised manuscript for spelling and grammatical errors.
3. Chemical structures for galantamine and memantine are missing in Table 1.
The structures were now introduced in Table 1.
4. Tables 2 and 3 seem to contain data from 2017, they should be updated to reflect notable failures in clinical trials from 2018 (i.e. Verubecestat).
We thank the reviewer for this comment and highlighted in the revised manuscript this failure: “In February 2018, APECS clinical trial was also discontinued and Merck no longer listed verubecestat in its research pipeline. Participants taking verubecestat scored worse than the placebo group on the cognitive test Alzheimer’s Disease Cooperative Study-Activities of Daily Living (ADCS-ADL), in a small but significant way. Further research in needed to understand the origin of this negative effect, if it is reversible and if can be associated with certain patient populations or stages of disease [90], [91]”. Moreover, we updated according AD pipeline to the end of 2018 Tables 2, 3, figure 1 and the texts associated: “However, besides the mentioned difficulties in the development of new therapies for AD, there are currently large research programs for the drug development of new anti-AD agents. In 2018, there were 112 agents in the AD pipeline, of which 23 in phase I, 63 in phase II and 26 in phase III. Across all stages, 63% are DMTs, 34% symptomatic agents for neuropsychiatric and behavioral changes, and 3% have undisclosed mechanism of action (MoA) [14]. Figure 1 resumes the MoA of the agents in clinical trials in 2018. Twenty five percent of the agents have a MoA amyloid related (12% immunotherapy and 14% small molecules), 14% anti-tau, 22% neurotransmitter based (symptomatic treatment agents), 18 % neuroprotection/ anti-inflammatory, 18% metabolic and 3% with other mechanisms [14]” and “Considering the 112 agents in the AD pipeline in 2018 [14] there are 29 with a mechanism of action amyloid related, 13 corresponding to immunotherapeutic agents (Table 2) and 16 to small molecules (Table 3) involved in phase I, II and III clinical trials.”.
Reviewer 2 Report
This is an excellent review, with an emphasis on chemistry of compounds and clinical trials.
Some minor points noted, which the authors are free to modify.
In introduction, mention that this review focuses mainly in small molecules, and not immunotherapy against amyloid, as each antibody trial was not discussed in detail.
Line 13: “Have not yet well succeeded”. Change to: “Have not been successful”
Table 1: Galantamine and memantine structures are not shown.
Figure 2: It’s not clear what the two vertical bars at the left of the figure mean.
Figure 5. It is not clear what delta, gamma, gamma prime cleavage refers to. Put references.
Figure 12: Addiction in several places should read addition.
Line 372: I suggest saying that the recently solved structure of gamma-secretase bound to C99 could help the development of potent and selective GS modulators.
Line 370: “deficiency”, change to “lack”.
Line 586: Don’t break the paragraph since it continues the idea about the same compound.
Line 819: “as” change to “such as”.
Line 143: Through the rest of the manuscript I didn’t see specific researchers named, so, as a question of style, I suggest removing the name Eric Karran, and just place the appropriate reference.
Author Response
This is an excellent review, with an emphasis on chemistry of compounds and clinical trials.
Some minor points noted, which the authors are free to modify. In introduction, mention that this review focuses mainly in small molecules, and not immunotherapy against amyloid, as each antibody trial was not discussed in detail.
Table 1: Galantamine and memantine structures are not shown.
The structures were now introduced in Table 1.
Figure 2: It’s not clear what the two vertical bars at the left of the figure mean.
We thank the reviewer and Figure 2 was updated (removal of the two vertical bars on the left).
Figure 5. It is not clear what delta, gamma, gamma prime cleavage refers to. Put references.
We thank the reviewer and Figure 5 was updated to clarify epsilon, delta, gamma and gamma prime cleavage. Also, legends were added and associated in the text as following: “The ε cleavage (1) releases the APP intracellular domain (AICD) and produces Aβ49 or Aβ48 [28], [31]. Then, the carboxypeptidase cleavages ζ (2) and γ (3 and 4) progressively trims these longer Aβ forms in both Aβ40 and Aβ42 [31], [32].”
Line 13: “Have not yet well succeeded”. Change to: “Have not been successful” Figure 12: Addiction in several places should read addition. Line 370: “deficiency”, change to “lack”. Line 586: Don’t break the paragraph since it continues the idea about the same compound. Line 819: “as” change to “such as”.
We thank the reviewer for identifying these lapses. We modified accordingly.
Line 372: I suggest saying that the recently solved structure of gamma-secretase bound to C99 could help the development of potent and selective GS modulators.
We agree and added the following sentence: “The recently solved structure of gamma-secretase bound to the C99 fragment could help the development of potent and selective GSM compounds [60].”
Line 143: Through the rest of the manuscript I didn’t see specific researchers named, so, as a question of style, I suggest removing the name Eric Karran, and just place the appropriate reference.
Throughout the manuscript, every time a specific research work is mentioned, the first author is nominated (Lines 145, 243, 254 and 256 of the updated manuscript).